# Adiponectin preserves metabolic fitness during aging

Na Li[1,2†], Shangang Zhao[1†], Zhuzhen Zhang[1], Yi Zhu[1], Christy M Gliniak[1], Lavanya Vishvanath[1], Yu A An[1], May-yun Wang[1], Yingfeng Deng[1], Qingzhang Zhu[1], Bo Shan[1], Amber Sherwood[3], Toshiharu Onodera[1], Orhan K Oz[3], Ruth Gordillo[1], Rana K Gupta[1], Ming Liu[2], Tamas L Horvath[4], Vishwa Deep Dixit[4,5], Philipp E Scherer[1,6]*

[1]Touchstone Diabetes Center, Department of Internal Medicine, The University of Texas Southwestern Medical Center, Dallas, United States; [2]Department of Endocrinology and Metabolism, Tianjin Medical University General Hospital, Tianjin, China; [3]Department of Radiology, University of Texas Southwestern Medical Center, Dallas, United States; [4]Department of Comparative Medicine and Immunobiology, Yale School of Medicine, New Haven, United States; [5]Yale Center for Research on Aging, Yale School of Medicine, New Haven, United States; [6]Department of Cell Biology, The University of Texas Southwestern Medical Center, Dallas, United States

*For correspondence:
philipp.scherer@utsouthwestern.edu

[†]These authors contributed equally to this work

Competing interests: The authors declare that no competing interests exist.

**Abstract** Adiponectin is essential for the regulation of tissue substrate utilization and systemic insulin sensitivity. Clinical studies have suggested a positive association of circulating adiponectin with healthspan and lifespan. However, the direct effects of adiponectin on promoting healthspan and lifespan remain unexplored. Here, we are using an adiponectin null mouse and a transgenic adiponectin overexpression model. We directly assessed the effects of circulating adiponectin on the aging process and found that adiponectin null mice display exacerbated age-related glucose and lipid metabolism disorders. Moreover, adiponectin null mice have a significantly shortened lifespan on both chow and high-fat diet. In contrast, a transgenic mouse model with elevated circulating adiponectin levels has a dramatically improved systemic insulin sensitivity, reduced age-related tissue inflammation and fibrosis, and a prolonged healthspan and median lifespan. These results support a role of adiponectin as an essential regulator for healthspan and lifespan.

## Introduction

Healthspan and lifespan are intimately linked. Improving healthspan should help enhance the overall quality of life for an aging population, and possibly even extend lifespan (*Crimmins, 2015*; *Piskovatska et al., 2019*). According to current estimates, by 2050, the number of older adults in the USA above the age of 65 years is expected to double, rising from 40.2 million to approximately 88 million (https://www.cdc.gov/nchs/products/databriefs/db106.htm). In the USA, the average lifespan is around 79.3 years, while the estimated healthspan is only 67.3 years, indicating that the individuals will on average live up to 20% of their lives in an unhealthy state (*Olshansky, 2018*). Moreover, 35–40% of adults aged 65 and above are obese. Given both aging and obesity are independent risk factors for chronic diseases, it is important to further determine how the confluence of adiposity and aging impacts healthspan and lifespan. The primary health problems associated with elderly individuals are obesity and associated metabolic disorders, including insulin resistance, type 2 diabetes, non-alcoholic fatty liver disease, hypertension, cardiovascular disease, and many types of cancers. These diseases are global public health problems, significantly accelerating the aging

process, and severely decreasing the quality of life and overall life expectancy (*Jura and Kozak, 2016*). Thus, increasing healthspan by prolonging a disease-free period of elderly individuals may be equally important as increasing lifespan. Simple strategies, such as caloric restriction, or pharmacological interventions, such as metformin or rapamycin treatment, can promote both healthspan and lifespan in mice (*Bhullar and Hubbard, 2015*; *Bitto et al., 2016*; *Martin-Montalvo et al., 2013*; *Minor et al., 2010*). However, the effectiveness of such an approach in humans still awaits confirmation. The search for novel and effective strategies to extend these processes is still one of the major goals of geroscience research.

Adiponectin was one of the earliest adipokines described (*Scherer et al., 1995*). Since its discovery, significant efforts have been made to study its regulation, biogenesis, and physiological effects. As an excellent biomarker for mature adipocytes, circulating adiponectin levels are inversely correlated with fat mass, distinguishing it from most of the other adipokines, including leptin (*Hu et al., 1996*). Adiponectin exerts pleiotropic effects, including improving glucose tolerance, increasing insulin sensitivity, enhancing lipid clearance, and reducing systemic inflammation and tissue fibrosis (*Scherer, 2006*). Our previous studies have indicated that a lack of adiponectin in mice leads to glucose intolerance and hyperlipidemia (*Nawrocki et al., 2006*; *Xia et al., 2018*). Conversely, increasing adiponectin levels in an adiponectin transgenic (ΔGly) mouse model greatly improves metabolic homeostasis and produces a metabolically healthy obese phenotype (*Combs et al., 2004*; *Kim et al., 2007*). Similarly, chronic administration of adiponectin ameliorates glucose intolerance and enhances insulin sensitivity in both type 1 and 2 diabetic mice (*Berg et al., 2001*). These observations fully support the favorable effects of adiponectin in promoting metabolic health.

Most of the previously published literature focuses on beneficial effects of adiponectin in younger mice or diet-induced obese mice within less than 20 weeks of a high-fat diet (HFD) challenge. Whether similar beneficial effects could be observed in aging mice (older than 100 weeks) remains unexplored. Beyond its possible role in healthspan, some human genetics studies have implicated adiponectin as a longevity gene (*Atzmon et al., 2008*). One potential mechanism of particular interest, with robust effects on elevating circulating adiponectin levels, is the starvation hormone fibroblast growth factor-21. It extends lifespan in both male and female mice (*Holland et al., 2013*). Similarly, thiazolidinediones, agonists of the peroxisome proliferator-activated receptor γ (PPARγ), also significantly increase circulating adiponectin levels and ameliorate aged-related tissue function decline (*Viljoen and Sinclair, 2009*; *Yu et al., 2002*). In addition, female mice harbor higher circulating adiponectin levels and live longer compared to male mice (*Gehrand et al., 2016*). All these observations point to a positive correlation between high circulating adiponectin and longevity and implicate adiponectin as a novel circulating hormone that may directly promote both healthspan and lifespan in mice. To test this hypothesis, we used our established mouse models of adiponectin overexpression and complete absence of adiponectin and assessed the effect of circulating adiponectin on the aging process. Our results reveal that adiponectin null (APN-KO) mice have a significantly reduced healthspan and lifespan, while ΔGly mice have a significantly prolonged healthspan.

## Results

### Altered adiponectin levels in APN-KO and adiponectin overexpressing transgenic mice

Male APN-KO mice (*Nawrocki et al., 2006*) and ΔGly mice (*Combs et al., 2004*) were used for this study. The initial number of mice for each group in the study and a detailed scheme of the phenotypic assessments performed are outlined in *Figure 1—figure supplement 1A and B*. APN-KO were challenged with chow diet (NCD) or HFD. ΔGly mice were challenged with NCD. Consistent with expectations, serum adiponectin was absent in APN-KO mice (*Figure 1—figure supplement 1C*). For ΔGly mice, circulating adiponectin levels were increased by 50% (*Figure 1—figure supplement 1C*). All these observations indicate that our loss and gain of function mouse models indeed alter circulating adiponectin levels effectively as expected.

### Deletion of adiponectin in aged mice shortens lifespan on HFD

Given that the loss of adiponectin leads to impaired glucose tolerance and lipid clearance, we wanted to test whether these mice have a shortened lifespan. A cohort of APN-KO and wild-

type (WT) mice was used to measure the lifespan. The survival curves for APN-KO reveal a statistically significant shortened lifespan compared to WT control both in the NCD cohort (*Figure 1A*) and in the HFD cohort (*Figure 1B*). Thus, loss of adiponectin in mice accelerates the aging process and shortens lifespan.

### Loss of adiponectin impairs glucose and lipid homeostasis during aging

Glucose intolerance is a hallmark of the aging process (*DeFronzo, 1981*). Compared to WT mice, APN-KO mice did not show any striking difference in body weight at middle and advanced age, both on NCD and HFD (*Figure 2A and B*). However, APN-KO mice exhibit a slight reduction in fat mass at advanced age (*Figure 2—figure supplement 1A*). The reduction of fat mass seems to be due mainly to the loss of subcutaneous fat (*Figure 2—figure supplement 1A*). We also found that adiponectin-deficient mice have increased bone content and higher bone mineral density during HFD feeding but normal values for NCD cohort (*Figure 2—figure supplement 1B and C*). We examined glucose homeostasis in aged mice (100 weeks for the HFD cohort and 140 weeks for the NCD cohort). In accordance with previous metabolic studies of young APN-KO mice, differences in glucose tolerance were marginal in mice fed standard NCD (*Figure 2C*). APN-KO mice fed HFD, in contrast, exhibited significantly higher glucose excursions during an oral glucose tolerance test (OGTT) (*Figure 2D*) reflecting impaired glucose tolerance. However, no significant difference in plasma insulin level was observed during the OGTT at the different time points (*Figure 2—figure supplement 1D*). This indicates that APN-KO mice are more susceptible to diet-induced glucose intolerance.

To elucidate the effects of adiponectin on lipid metabolism of aged mice, we performed a triglyceride (TG) clearance test by gavaging the WT and APN-KO mice with 20% intralipid. Triacylglycerol levels in both NCD- and HFD-fed APN-KO mice peaked at higher levels and showed a slower

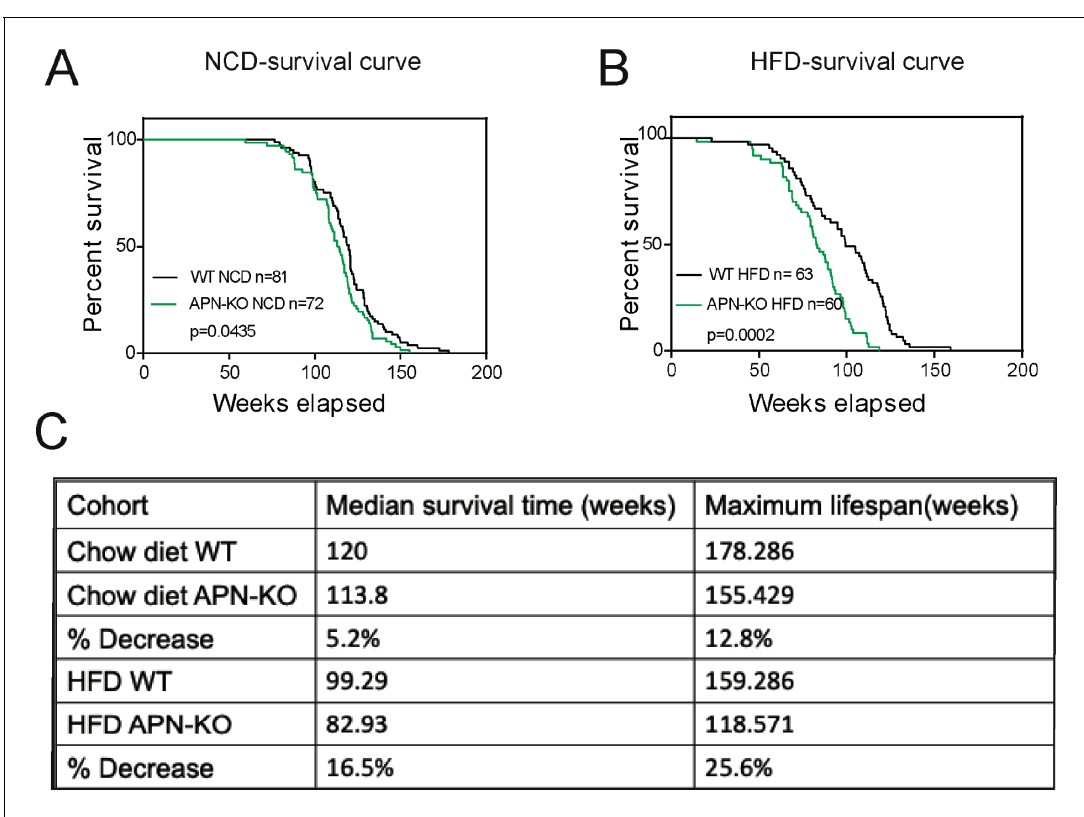

| Cohort | Median survival time (weeks) | Maximum lifespan(weeks) |
|---|---|---|
| Chow diet WT | 120 | 178.286 |
| Chow diet APN-KO | 113.8 | 155.429 |
| % Decrease | 5.2% | 12.8% |
| HFD WT | 99.29 | 159.286 |
| HFD APN-KO | 82.93 | 118.571 |
| % Decrease | 16.5% | 25.6% |

**Figure 1.** Lack of adiponectin (APN) in aging mice shortens lifespan. (**A**) Kaplan–Meier survival curves for wild-type (WT) and adiponectin null (APN-KO) mice on chow diet. (**B**) Kaplan–Meier survival curves for WT and APN-KO mice on high-fat diet (HFD). (**C**) Median survival time and maximum lifespan for each cohort. n denotes the number of mice per group.p-Values were determined by log-rank (Mantel–Cox) test.

The online version of this article includes the following figure supplement(s) for figure 1:

**Figure supplement 1.** Mouse models used for longevity studies: adiponectin null (APN-KO) mice and adiponectin transgenic (ΔGly) mice.

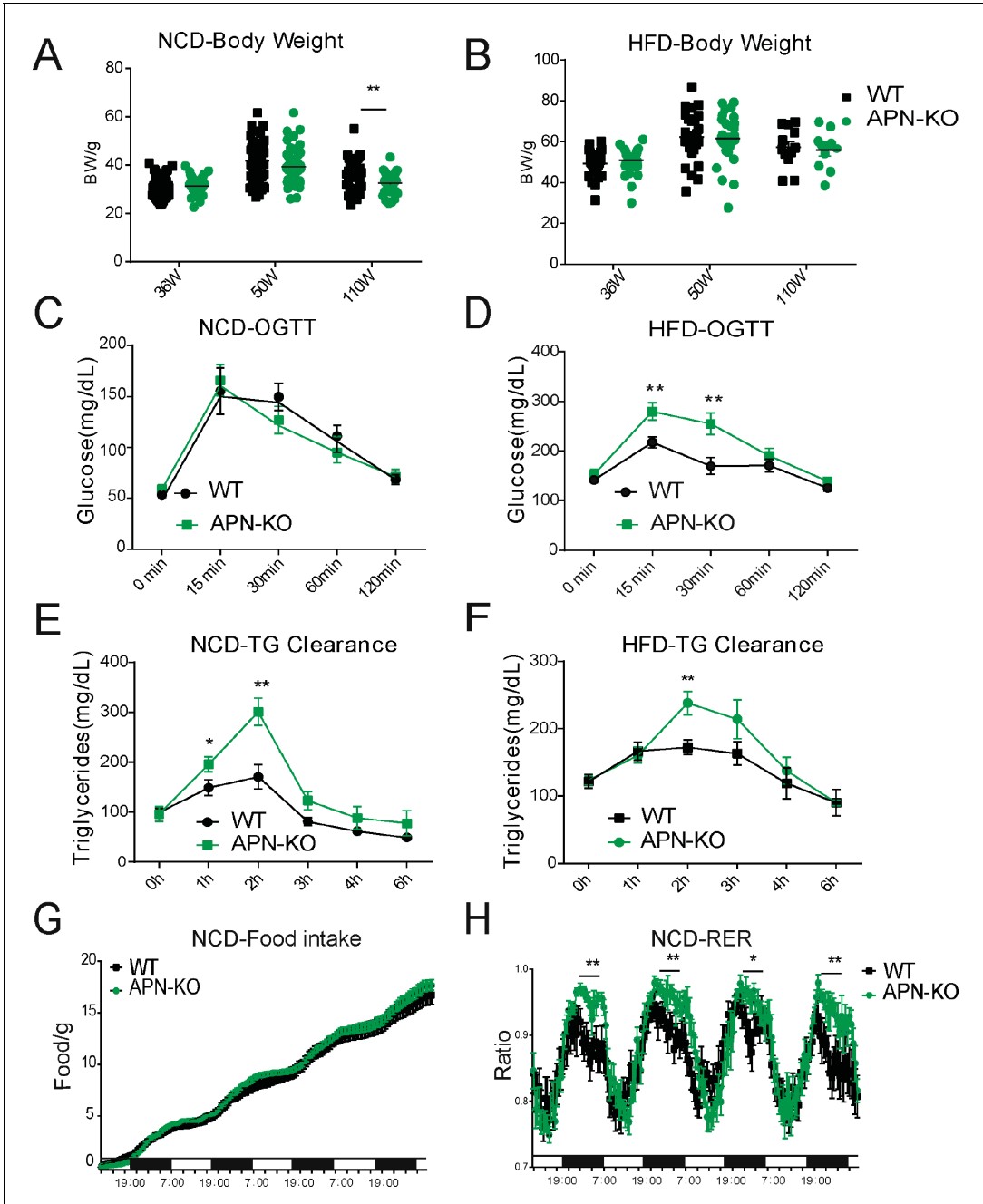

**Figure 2.** Lack of adiponectin (APN) in aging mice worsens glucose and lipid homeostasis. (A) Body weights during aging processes for wild-type (WT) and adiponectin null (APN-KO) mice fed on chow diet. (B) Body weights during aging processes for WT and APN-KO mice fed on high-fat diet (HFD). (C) An oral glucose tolerance test (OGTT) (2 g kg$^{-1}$ body weight; single gavage) on chow diet-feeding WT and APN-KO mice at 110 weeks old (n = 7 per group). (D) An OGTT (1.25 g kg$^{-1}$ body weight; single gavage) on HFD-feeding WT and APN-KO mice at 85-weeks old (n = 8 for WT, n = 7 for APN-KO mice). (E) Triglyceride (TG) clearance test (20% intralipid; 15 μl g$^{-1}$ body weight; single gavage) in chow diet-feeding WT and APN-KO mice at 110 weeks old (n = 9 for WT, n = 10 for APN-KO mice). (F) TG clearance test (20% intralipid; 15 μl g$^{-1}$ body weight; single gavage) in HFD-feeding WT and APN-KO mice at 85 weeks old (n = 8 per group). (G) Metabolic cage analyses showing food intake for chow diet-feeding WT in APN-KO mice at 110 weeks old (n = 8 for WT, n = 7 for APN-KO mice). Data are mean ± SEM. Student's *t* test: *p<0.05, **p<0.01, ***p<0.001 for WT vs. APN-KO. (H) Metabolic cage analyses showing respiratory exchange rate (RER) chow diet-feeding WT and APN-KO mice at 110 weeks old (n = 8 for WT, n = 7 for APN-KO mice). Data are mean ± SEM. Student's *t* test: *p<0.05, **p<0.01, ***p<0.001 for WT vs. APN-KO.

The online version of this article includes the following figure supplement(s) for figure 2:

**Figure supplement 1.** Body composition of wild-type (WT) mice and adiponectin null (APN-KO) mice.

clearance of lipids from plasma (*Figure 2E and F*). This highlights a prevailing impaired lipid clearance in APN-KO mice. Furthermore, although APN-KO and WT mice consume comparable amounts of diet (*Figure 2G*), indirect calorimetric studies show that APN-KO mice had a significantly higher respiratory exchange ratio (*Figure 2H*), indicative of carbohydrate being a more predominant fuel source in the absence of adiponectin. Combined, these results suggest that adiponectin is necessary to maintain proper lipid homeostasis. Lack of adiponectin prompts glucose metabolism to be more prevalent.

## Deletion of adiponectin in aged mice exacerbates tissue functional decline

The aging process is associated with gradual decline and deterioration of functional properties at the tissue level. In aging adipose tissue, this is manifest as expansion of B cells in fat-associated lymphoid clusters (*Camell et al., 2019*), enrichment of senescent-like pro-inflammatory macrophages, and loss of tissue protective macrophage subsets that drive inflammaging and compromise glucose and lipid metabolism (*Camell et al., 2017*; *Lumeng et al., 2011*; *Tchkonia et al., 2010*). In the liver and kidney, dysfunction is usually apparent as overexpression of extracellular matrix (ECM) protein constituents, such as collagen and the resulting increased fibrosis (*Kim et al., 2016*). We examined whether the deletion of APN will affect the function of these major organs. We collected adipose tissue, kidney, and liver from separate aging cohorts of young (20 weeks) and old (100 weeks for HFD cohort and 140 weeks for NCD cohort) mice. Compared to WT mice, APN-KO mice did not show significant morphological differences in adipocytes in both young and aged mice. However, the epididymal fat of APN-KO mice fed either HFD or NCD show increased pro-inflammatory-like macrophages in the aged mice, as demonstrated by a prominent signal for the macrophage marker Mac2 (*Figure 3A and B*). Consistent with the immunohistochemical staining, the expression levels of inflammatory markers were drastically increased in gonadal white adipose tissue of APN-KO mice, both on NCD and on HFD (*Figure 4A*). This demonstrates that the loss of adiponectin accelerates adipose tissue inflammation, a characteristic marker of the increased aging process.

We also examined the age-related decline of health parameters in two other vital organs, kidney and liver. Even during normal aging, the kidney develops age-related structural changes and displays functional declines, including nephrosclerosis, loss of renal mass, or compensatory hypertrophy of the remaining nephrons, with a corresponding decrease in glomerular filtration rate and renal blood flow (*Weinstein and Anderson, 2010*). Clinical studies have demonstrated that adiponectin is elevated in patients with chronic kidney disease, suggesting a possible compensatory upregulation to alleviate further renal injury (*Christou and Kiortsis, 2014*). Morphologically, APN-KO mice fed either the HFD or the NCD show more severe interstitial and periglomerular fibrosis. Compared to aged WT mice, the glomeruli in aged APN-KO mice have collapsed tufts, accompanied by hypertrophic Bowman's capsules (*Figure 3C*). Meanwhile, aged APN-KO mice exhibited a significant increase in kidney weight as compared with aged WT mice (*Figure 2—figure supplement 1E*). To determine the cause of this severe glomerular and tubulointerstitial damage in APN-KO mice, we investigated the glomerular infiltration with macrophages. Immunohistochemical staining with Mac2 antibodies reveals a significant increase in Mac2 positive intraglomerular signal in the old mice which is vastly more abundant in the APN-KO mice fed the HFD (*Figure 3D*). Markers of kidney inflammation, such as TNFα, IL1β, IL10, F4/80, MCP-1, and CRP, were assessed in total kidney tissues, isolated from both APN-KO and control mice. Across the board, the expression levels of inflammation markers were dramatically increased in adiponectin-deficient mice (*Figure 4B*).

Aging increases the susceptibility of various liver diseases as well, responsible for a deteriorated quality of life in the elderly and increasing mortality rate. Several studies suggest that hypoadiponectinemia predicts liver fibrosis and accelerates hepatic tumor formation (*Park et al., 2015*). Thus, we explored whether the lack of adiponectin may exacerbate age-induced dysfunction and dysmorphology of the liver. Unlike other diet-induced obese mouse models, we did not find any enhanced lipid droplet accumulation in the livers of APN-KO mice compared to WT mice upon short-term and long-term exposure to HFD treatment (*Figure 3E*). However, we found many inflammatory infiltrates in the livers of APN-KO mice on HFD diet. The expression of inflammatory markers is significantly increased in aged APN-KO mice fed on HFD and NCD (*Figure 4D* and *Figure 4—figure supplement 1A*), indicative of increased inflammation in the liver. Meanwhile, we found corticosteroids, key systemic anti-inflammatory factors, were upregulated in APN-KO mice compared to control mice,

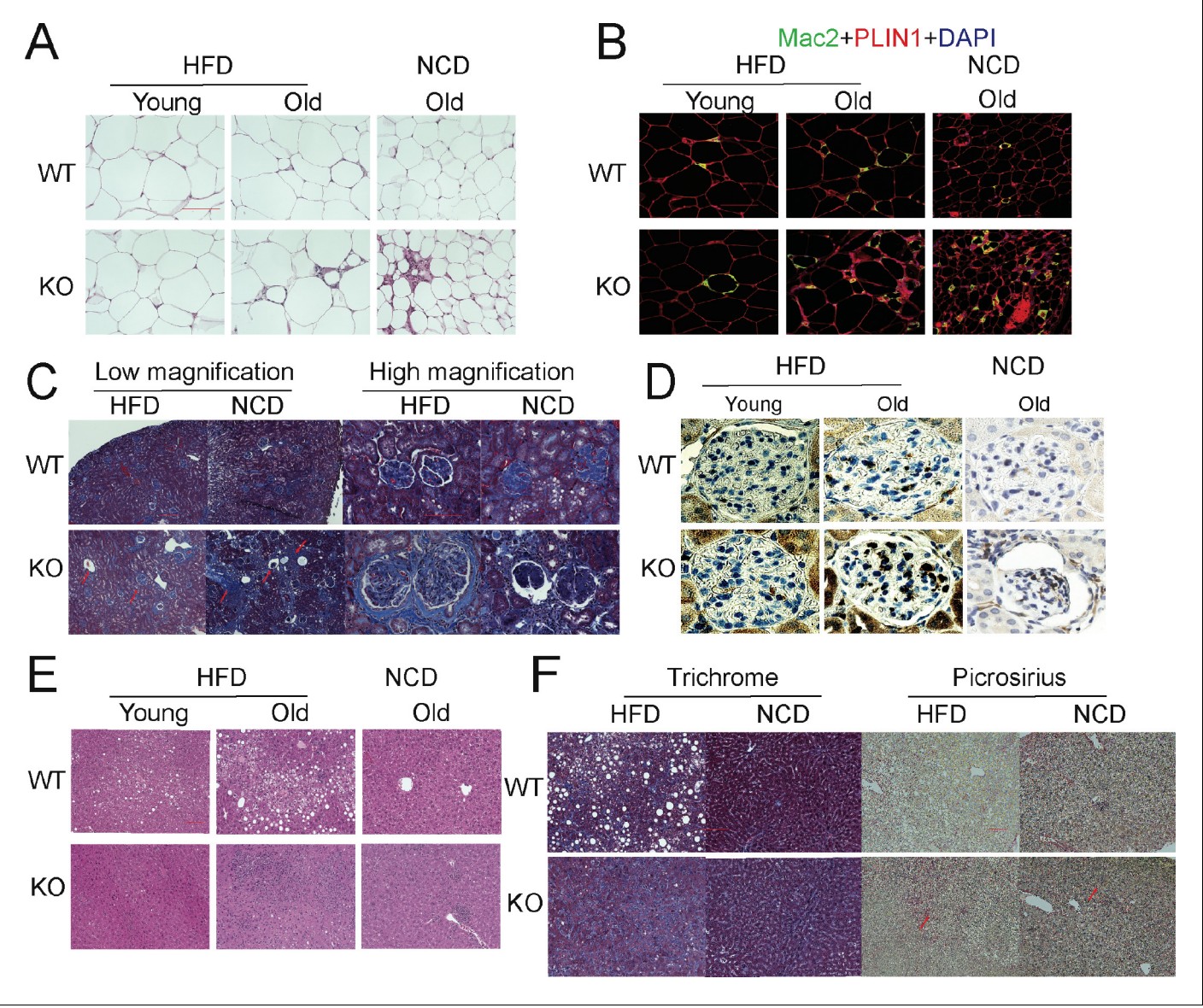

**Figure 3.** Deletion of adiponectin (APN) in aged mice exacerbates functional decline. (**A**) H&E staining of an Epi fat depot of 20-week-old and 100-week-old wild-type (WT) and adiponectin null (APN-KO) mice fed on high-fat diet (HFD) or 140-week-old WT and APN-KO mice on chow diet. (**B**) Mac2 staining of an Epi fat depot of 20-week-old and 100-week-old WT and APN-KO mice fed on HFD or 140-week-old WT and APN-KO mice on chow diet. (**C**) Trichrome staining of kidney sections reveals severe interstitial and periglomerular fibrosis in 110-week-old APN-KO mice fed on HFD and 140-week-old APN-KO mice fed on chow diet. Collapsed tufts are seen inside widened Bowman's capsules forming glomerular cysts (red arrow). (**D**) Mac2 staining of kidney sections of 20-week-old and 100-week-old WT and APN-KO mice fed on HFD or chow diet. (**E**) H&E staining of liver of 20-week-old and 100-week-old WT and APN-KO mice fed on HFD, 140-week-old WT and APN-KO mice on chow diet. Note the extensive inflammatory cell infiltrates in the liver of the aged APN-KO mice fed on HFD. (**F**) Trichrome and Picrosirius stains of liver sections from 100-week-old WT and APN-KO mice fed on HFD or 140-week-old WT and APN-KO mice on chow diet examine liver fibrosis. Bar, 100 μm. Data are mean ± SEM. Student's $t$ test: *p<0.05, **p<0.01, ***p<0.001 for WT vs. APN-KO.

presumably as a compensatory response (*Figure 4—figure supplement 1B*). To further assess the origin of these inflammatory markers, we have isolated macrophages from livers of WT and APN-KO mice and sorted out Kupffer cells and monocyte-derived macrophages. We found that the total number of macrophages in APN-KO mice is doubled (*Figure 4C*). Moreover, compared to WT mice, Kupffer cells and monocyte-derived macrophages both increased in APN-KO mice (*Figure 4C*), and Kupffer cells are the major source of the inflammatory response (*Figure 4C*). Hepatic stellate cells are the major cell types that produce collagen in response to a liver insult. In the APN-KO mice,

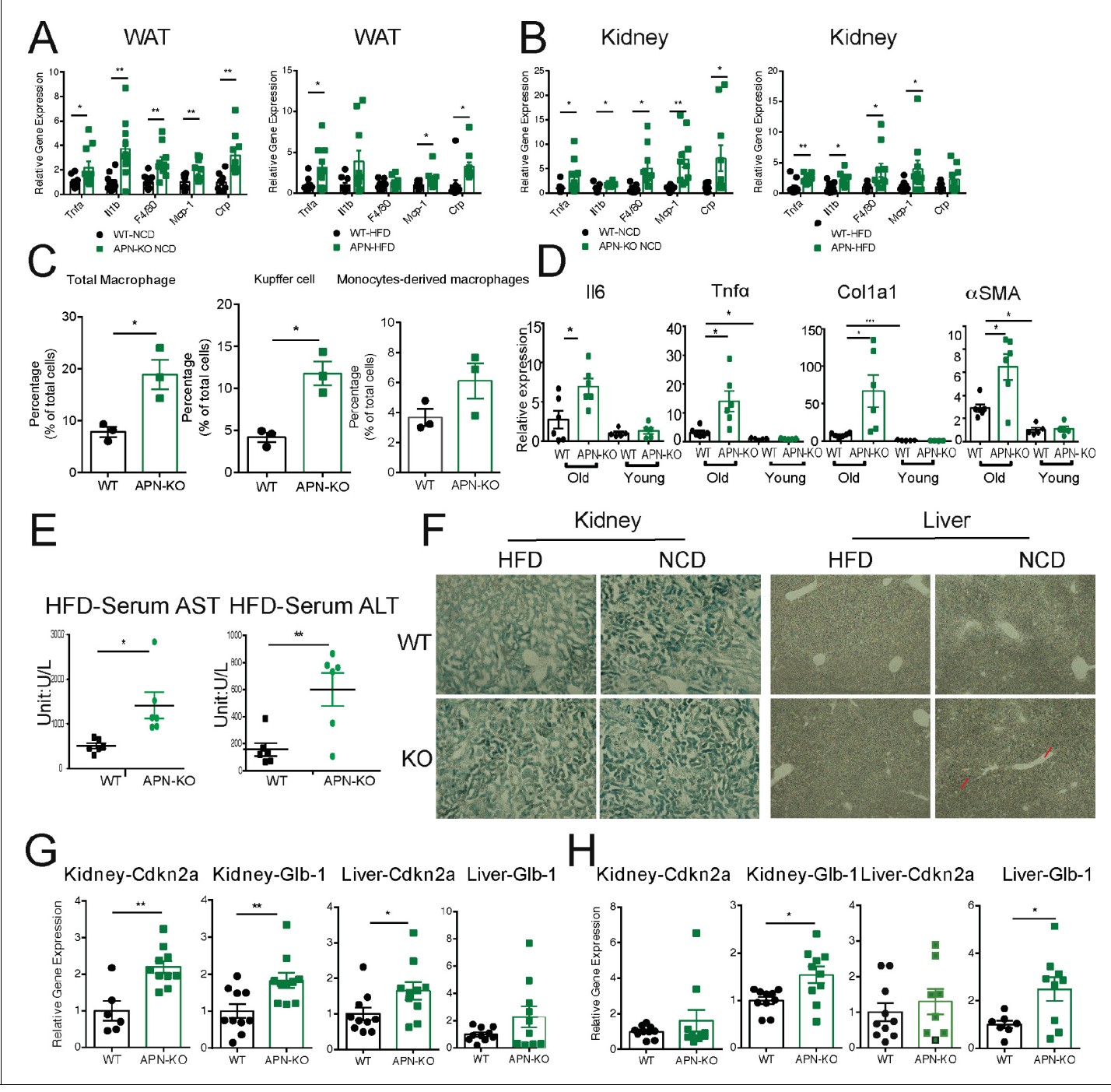

**Figure 4.** Absence of adiponectin (APN) in aged mice exacerbates inflammation and accelerates aging. (**A**) Expression of inflammatory markers in epididymal fat depots of 140-week-old wild-type (WT) and adiponectin null (APN-KO) mice fed on chow diet and 100-week-old WT and APN-KO fed on HFD(n = 10 per group). (**B**) Expression of inflammatory markers in kidneys of 140-week-old WT and APN-KO mice fed on chow diet and 100-week-old WT and APN-KO HFD (n = 8–10 per group). (**C**) FACS analysis of percentages of total macrophages, Kupffer cells, and monocytes-derived macrophages isolated from 100-week-old WT and APN-KO mice fed on HFD (n = 3 per group). (**D**) Expression of inflammatory and fibrosis markers in liver tissues of 20-week-old and 100-week-old WT and APN-KO mice fed on HFD (n = 5 per group of young cohorts, n = 6 per group of aged cohorts). (**E**) Serum AST and ALT activities in 100-week-old WT and APN-KO mice fed on HFD (n = 6 per group). (**F**) β-Galactosidase staining of kidney and liver sections from 100-week-old WT and APN-KO mice fed on HFD or 140-week-old WT and APN-KO mice on chow diet examines cellular senescence. (**G**) Expression of senescence biomarkers in kidneys and livers of 140-week-old WT (n = 6 or 10) and APN-KO mice fed on chow diet (n = 10). (**H**) Expression of senescence biomarkers in kidneys and livers of 100-week-old WT (n = 7–10) and APN-KO mice fed on HFD (n = 8–10). Bar, 100 μm. Data are mean ± SEM. Student's *t* test: *p<0.05, **p<0.01, ***p<0.001 for WT vs. APN-KO.

*Figure 4 continued on next page*

Figure 4 continued

The online version of this article includes the following figure supplement(s) for figure 4:

**Figure supplement 1.** Adiponectin deficiency in aged mice exacerbates inflammation and accelerates aging.

significant increases in liver inflammation were observed, in part due to elevated total macrophage numbers and enhanced macrophage activity (*Figure 4—figure supplement 1C*), thereby imposing a strong insult to the livers, resulting in hepatic stellate cell activation and liver fibrosis. Moreover, trichrome and Picrosirius red stains highlighting ECM components reveal increased hepatic fibrosis in old APN-KO mice on NCD that was even more evident under HFD conditions (*Figure 3F*). Mirroring these histological findings, the expression levels of liver fibrosis markers, such as Col1α1 and αSMA, are strikingly increased in older HFD and NCD fed APN-KO mice (*Figure 4D* and *Figure 4—figure supplement 1A*). Liver damage was further confirmed by elevated serum AST and ALT levels in HFD fed APN-KO mice compared with control mice (*Figure 4E*). Senescent cells increasingly accumulate within tissues over the course of aging. Livers and kidneys of aged APN-KO mice display a higher percentage of senescent cells (as judged by senescence-associated β-galactosidase staining) compared to control mice (*Figure 4F*). Moreover, Cdkn2a and Glb1, as highly distinctive senescence markers, are also upregulated in kidneys and livers of APN-KO mice, both on NCD and on HFD (*Figure 4G and H*). The accumulation of these senescent cells over time may contribute to progressive kidney and liver failure observed. All of these observations support a model that suggests that adiponectin plays an essential role in maintaining normal liver function during the aging process.

Upon comparing young WT vs. APN-KO mice (20 weeks) that were exposed for 8 weeks to HFD, no genotype-specific differences were observed in the kidney and the liver. This therefore indicates that the pathological changes in older APN-KO mice genuinely reflect age-related chronic changes rather than simple developmental differences that would be apparent in the young mice as well. These findings clearly indicate that the lack of adiponectin during aging exacerbates liver and renal damage, at least in part through pro-inflammatory mechanisms.

## Increasing adiponectin protects mice from aged induced metabolic dysfunction

Clinically, adiponectin levels are significantly higher in centenarians and in some of their offspring, suggesting that adiponectin may be a key driver to promote healthspan and lifespan. As the elimination of adiponectin shortens healthspan and lifespan, we wondered whether increasing adiponectin by our previously established transgenic mouse model (that we refer to as the 'ΔGly mouse') could promote both healthspan and lifespan. A large cohort of WT and ΔGly mice was placed on NCD to assess their lifespan. After calculation, a median lifespan in control mice was around 117 weeks, while this value in ΔGly mouse has been extended to 128 weeks (9% extension), indicating that increasing circulating adiponectin prolongs median lifespan. However, the maximum lifespan is comparable in control and ΔGly mice, as the overall survival curves were not different by log-rank test (*Figure 5A*).

Besides its positive effects in prolonging median lifespan, we determined if increasing adiponectin levels may have beneficial effects in extending healthspan. Previous studies indicated that increasing adiponectin levels results in improved glucose and lipid profiles in younger mice (*Berg et al., 2001*). However, whether these beneficial effects of adiponectin carry to older age has not been assessed. When fed with an NCD, ΔGly mice show a similar body weight during lifespan, compared to littermate controls (*Figure 5B*). Then, we measured fasting glycemia, insulin, and insulin-like growth factor 1 (IGF-1) (*Figure 5C*). Under 16 hr fasted conditions, ΔGly mice have a significantly lower fasting glycemia, accompanied by a robust reduction in plasma insulin. Moreover, a reduction in circulating IGF-1 levels is observed in ΔGly mice. Lower IGF-1 levels are thought to play a key role as a mediator of healthspan and lifespan extension (*Bartke et al., 2003*). To test whether the improvements in systemic insulin sensitivity are also associated with improvements at the level of the pancreatic β cell, we performed H&E staining on pancreatic sections. Consistent with the reduced demand on islets to produce and release insulin in a more insulin-sensitive environment, the average islet size was considerably reduced by adiponectin overexpression, with islet structural integrity fully preserved (*Figure 5D*). Immunohistochemical analysis of islets exhibits a normal

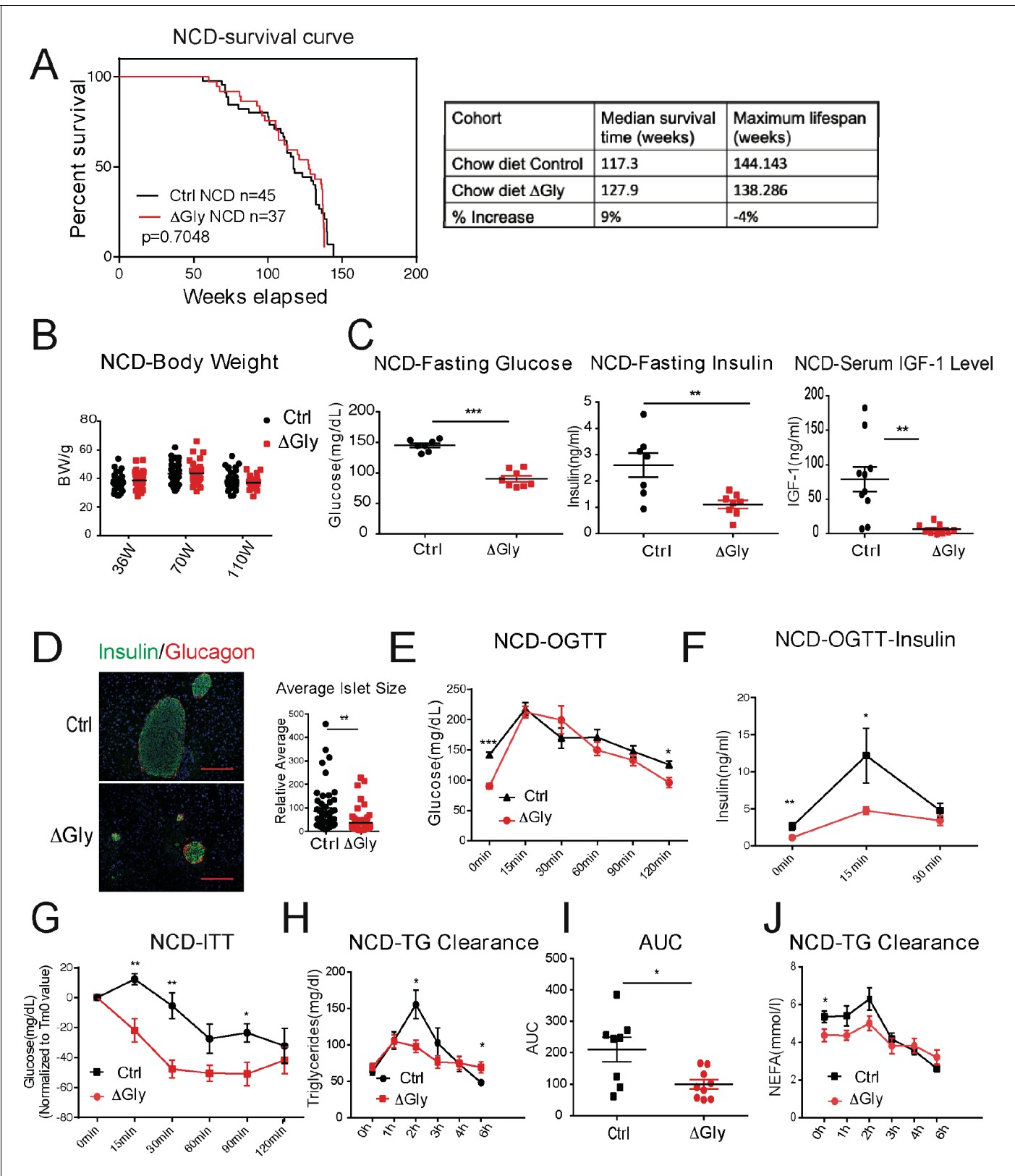

**Figure 5.** Increasing adiponectin protects against aging-induced metabolic disturbance. (**A**) Kaplan–Meier survival curves for controls and ΔGly mice on chow diet. Median survival time and maximum lifespan for each cohort. n denotes the number of mice per group. p-Values were determined by log-rank (Mantel–Cox) test. (**B**) Body weights during aging processes for controls and ΔGly mice fed on chow diet. (**C**) Systemic glucose, insulin, and insulin-
*Figure 5 continued on next page*

*Figure 5 continued*

like growth factor 1 (IGF-1) levels in 50-week-old controls and ΔGly mice after fasting 16 hr. (**D**) Insulin and glucagon IF staining of pancreases from controls and ΔGly mice at 140 weeks old (left). Right: Relative average islet size. (**E**) An oral glucose tolerance test (OGTT) (2 g kg$^{-1}$ body weight; single gavage) revealed marginally improved glucose tolerance in 50-week ΔGly compared with controls (n = 8 per group). (**F**) Serum insulin levels during glucose tolerance test performed in panel E (n = 7 for controls, n = 8 for ΔGly mice). (**G**) Insulin tolerance test (ITT) in controls and ΔGly mice at 50 weeks old (n = 8 per group). (**H**) Triglyceride (TG) clearance test in controls and ΔGly mice at 50 weeks old (n = 8 for controls, n = 9 for ΔGly mice). (**I**) Area under curve (AUC) calculated based on H. (**J**) Circulating free fatty acid (FFA) levels in controls and ΔGly mice at 50 weeks old during TG clearance performed in panel H (n = 8 for controls, n = 9 for ΔGly mice). Bar, 100 μm. Data are mean ± SEM. Student's *t* test: *p<0.05, **p<0.01, ***p<0.001 for controls vs. ΔGly.

composition with α cells (glucagon) and β cells (insulin) in ΔGly mice. During an OGTT, ΔGly mice displayed a much lower glucose excursion than littermates (*Figure 5E*). In addition, insulin levels in ΔGly mice were significantly lower in response to the glucose challenge, which further supports improved insulin sensitivity (*Figure 5F*). To confirm this, we performed insulin tolerance tests (ITTs). ΔGly mice show a significant increase in insulin sensitivity (*Figure 5G*), which is consistent with our results for the young mice. Moreover, when orally challenged with TGs, ΔGly mice display enhanced lipid clearance (*Figure 5H and I*), with correspondingly lower free fatty acid (FFA) values (*Figure 5J*). These data demonstrate that increasing adiponectin levels significantly promotes metabolic fitness in aged mice.

## Increasing adiponectin levels improves the age-related functional decline in tissues of aged mice

To probe tissue functional declines that might contribute to metabolic syndrome in the elderly, we evaluated the function of fat and liver in aged mice. Aging is associated with a redistribution of fat from the periphery to central fat deposition (*Kuk et al., 2009*). The redistribution and ectopic fat deposition with aging appear to accelerate onset of multiple age-related diseases. A histological examination of adipose tissue showed that ΔGly mice harbor much smaller adipocytes in subcutaneous and gonadal fat (*Figure 6A*) compared to controls at the age of 140 weeks. In agreement with the epididymal adipocyte size and fat mass, inflammation is potently suppressed in visceral fat tissues of ΔGly mice, as demonstrated by a significantly reduced Mac2 staining (*Figure 6B*). Moreover, it was quite apparent that visceral fat pad weight was reduced in ΔGly mice with a slightly increase in subcutaneous adipose tissue (*Figure 6C*). Aged WT mice revealed an unclear boundary in the hepatic lobule with lose cellular cytoplasm, while ΔGly mice entirely prevented lipid droplet accumulation and age-related deterioration of the morphology of the liver (*Figure 6D*). Meanwhile, considerably less collagen was deposited in the livers of ΔGly mice, as judged by the reduced ECM stain (*Figure 6E*). Furthermore, gene expression of inflammation and fibrosis markers in the livers were dramatically reduced, in parallel with decreased corticosterone levels in ΔGly mice compared to their littermates (*Figure 6F and G*). Combined, these findings strongly support that adiponectin promotes metabolic fitness, by maintaining a proper fat distribution and reducing adipose tissue inflammation, along with reducing inflammation and fibrosis in liver.

## Discussion

Based on data from clinical correlations as well as ample preclinical results, we appreciate that elevated levels of adiponectin are generally associated with an improved overall metabolic phenotype. Here, we systematically assessed the impact of adiponectin in the context of aging. Using APN-KO and adiponectin overexpressing mouse models, we have made the following observations: (1) The lack of adiponectin in mice curtails healthspan by impairing glucose and lipid homeostasis, and accelerating fibrogenesis in multiple tissues, resulting in reduced healthspan. (2) The lack of adiponectin in mice shortens lifespan both on NCD and HFD. (3) Increasing adiponectin levels in aged adiponectin overexpressing mice produces a healthy metabolic phenotype, with greatly increased glucose tolerance and insulin sensitivity, enhanced lipid clearance, lowered visceral fat, and potent protection from inflammation and fibrosis. (4) Adiponectin overexpressing mice on an NCD show a 9% increase in median lifespan. All these observations support that adiponectin is a vastly underestimated player in healthspan and lifespan.

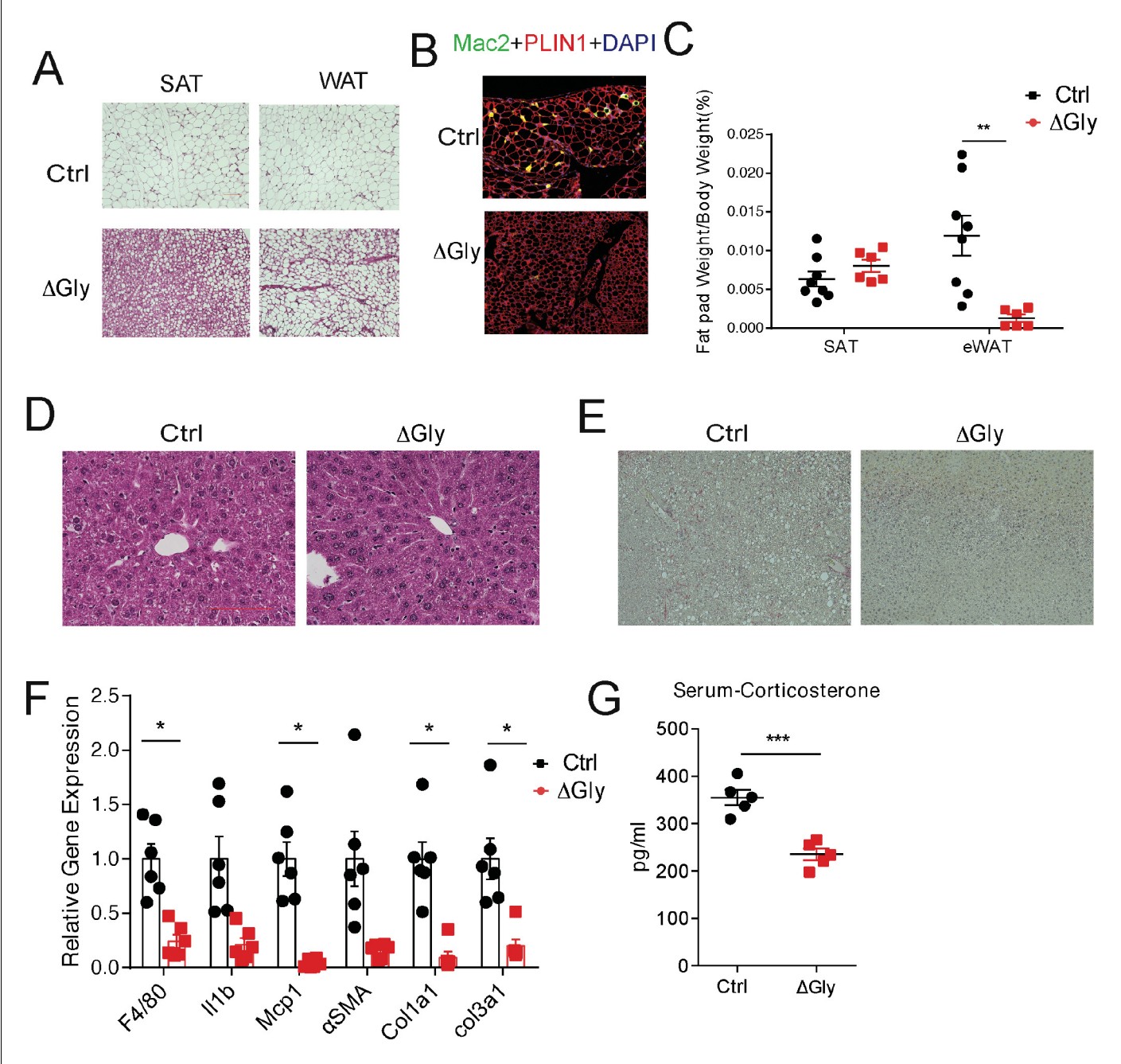

**Figure 6.** Old adiponectin overexpressing mice exhibit improved glucose and lipid homeostasis. (**A**) H&E staining of SubQ fat depot and Epi fat depot of 140-week-old controls and ΔGly mice fed on chow diet. (**B**) Mac2 staining of epididymal fat sections in 140-week-old controls and ΔGly mice. (**C**) Relative subcutaneous and visceral fat pad weights of 140-week-old controls and ΔGly mice fed on chow diet (n = 8 for controls, n = 6 for ΔGly mice). (**D**) H&E staining of liver from 140-week-old controls and ΔGly mice fed on chow diet. (**E**) Picrosirius red staining of livers from 140-week-old controls and ΔGly mice fed on chow diet. (**F**) Expression of inflammatory and fibrosis markers in liver of 140-week-old controls and ΔGly mice fed on chow diet (n = 8 for controls, n = 6 for ΔGly mice). (**G**) Serum corticosterone level in 140-week-old controls and ΔGly mice fed on chow diet (n = 5 per group). Bar, 100 µm. Data are mean ± SEM. Student's *t* test: *p<0.05, **p<0.01, ***p<0.001 for controls vs. ΔGly.

Regarding the effects of adiponectin on lifespan, it is quite clear that APN-KO mice with no circulating adiponectin significantly shorten their lifespan, while ΔGly mice with an increase in circulating adiponectin only slightly increase their median lifespan, despite the substantial improvement in glucose and insulin tolerance and significant reduction in tissue inflammation and fibrosis. Tissue

inflammation and fibrosis are suggested as one of the crucial contributing factors for lifespan. Consistent with this notion, increased liver inflammation and fibrosis in APN-KO mice results in a higher incidence of liver cancer. In HFD-fed aged WT mice, we observed a small number of mice with liver cancer (<10%). However, in HFD-fed aged APN-KO mice, the rate of liver cancer increased up to 60%. Thus, increased rate of liver cancer and its associated cachexia possibly account for the shortened lifespan in APN-KO mice. In addition, this observation also suggests that inflammation or fibrosis itself is not the actual cause of death for the mice, but a precursor to cellular transformation, particularly in the liver. This may help explain the reduced inflammation and fibrosis in ΔGly mice only slightly increases their lifespan, implicating additional factors, in addition to reduced inflammation and fibrosis, may contribute to the natural cause of death of these mice.

Adiponectin has potent anti-fibrotic effects in the liver by activating PPARγ pathways (*Shafiei et al., 2011*), which in turn diminishes the expression of pro-fibrotic genes. Despite having reduced levels of TG accumulation in the liver, the chronic lack of adiponectin dramatically exacerbates age-related liver fibrosis in parallel with disruption of liver function. In contrast, ΔGly mice are completely protected from diet- and aging-induced steatohepatitis and fibrosis, indicating of a crucial role of adiponectin in regulating liver inflammatory reactions and fibrosis. Hepatic stellate cells are the major cell types that produce collagen in response to liver insults (*Wang and Friedman, 2020*). In the APN-KO mice, significant increases in liver inflammation by elevated total macrophage numbers and enhanced macrophage activity impose a strong insult to liver. This results in hepatic stellate cell activation and liver fibrosis. In contrast, higher circulating levels of adiponectin, observed in ΔGly mice, greatly reduce liver inflammation, leading to reduced liver fibrosis. All these results suggest that local inflammation in livers is the major driver of liver fibrosis. Hence, the anti-inflammatory impact and potent anti-fibrotic actions of adiponectin make it a potent novel regulator of enhanced healthspan.

Obesity-associated chronic inflammation and insulin resistance are regarded as a pivotal risk factors for the development of several age-related pathological sequelae (*Huffman and Barzilai, 2009*; *Kanneganti and Dixit, 2012*). Improved metabolic homeostasis is positively associated with lifespan in humans and mice. Prolongevity interventions, caloric restriction, and long-lived Ames dwarf mice all have increased adiponectin expression (*Hill et al., 2016*). We found that adiponectin mimics to some extent the impact of caloric restriction on the reduction in inflammation and improved metabolic homeostasis. Reduction in adiposity is considered to be a hallmark of caloric restriction, which is an important component of its beneficial effects on metabolism. After long-term caloric restriction, aged mice display lower adiposity, smaller adipocytes, and improved insulin sensitivity (*Miller et al., 2017*). Strikingly, increases in adiponectin expression are detected in these smaller adipocytes after caloric restriction. Adipocyte hypertrophy is associated with cellular stress and obesity-associated metabolic complications. Due to the limited capacity to expand subcutaneous adipose tissue in aged populations, adipocyte hypertrophy also occurs in visceral fat, which is associated with lipid spillover in multiple tissues in aging and ectopic fat accumulation (*Tchkonia et al., 2013*). Hypertrophic adipocytes and an impaired redistribution of lipids exert a negative impact on insulin responsiveness, contributing to many metabolic diseases frequently observed in the elderly. Thus, metabolic disorders frequently go hand in hand with aging. Clinical studies have identified a strong inverse relationship between circulating adiponectin and insulin resistance in obese individuals (*Turer et al., 2011*). Our previous data suggested that adiponectin strongly suppresses hepatic gluconeogenesis and enhances fatty acid oxidation, thereby strongly contributing to an overall beneficial metabolic regulation (*Wang and Scherer, 2016*). Our aged ΔGly mice still have dramatically improved insulin sensitivity in parallel with reduced plasma insulin. All of this happens primarily due to an increase in adipocyte numbers in subcutaneous fat of ΔGly mice. In the absence of the protective effects of adiponectin, aged APN-KO mice exacerbate diet- and aging-induced glucose intolerance and lipid disorders. Moreover, ΔGly mice show improved insulin sensitivity in parallel with lower insulin and IGF-1 levels, and higher IGF-1 in APN-KO mice. Attenuated activation of the growth hormone–IGF-1 axis is also an integral component of the beneficial effects of caloric restriction, leading to prolonged healthspan and lifespan in rodents. Interestingly, elevated adiponectin is detected in all long-live mice: This includes the adipocyte-specific insulin receptor knockout mice, the Ames dwarfs (df/df), and GHRKO mice (*Blüher et al., 2002*; *Masternak et al., 2012*; *Wang et al., 2006*). The similarities across all these models suggest that an increase in adiponectin levels may be the common denominator driving longevity in all models mentioned.

Combined, our studies, along with previous reports, demonstrate that adipose tissue plays a vital role in the aging process. In aging, dysfunctional fat tissue leads to ectopic fat deposition, lipodystrophic adipocytes, and subcutaneous fat loss, thereby contributing to increased systemic inflammation, metabolic disturbances, and functional declines in other organs. However, healthy fat pads have characteristic features not only in terms of the quantity, but more importantly, by the quality of adipose tissue (*Kusminski et al., 2012*). Adiponectin is a key player maintaining glucose and lipid homeostasis on the basis of its lipid-storing capacity and its ability to communicate with other organs. Thus, overexpressing adiponectin results in the healthy expansion of subcutaneous adipose tissue, a reduction of visceral fat and an improvement in inflammation and fibrosis in the liver, all of which greatly alleviates metabolic disturbances and protects against tissue functional decline during the aging process. In contrast, adiponectin deficiency increases susceptibility to metabolic diseases in the elderly. Impaired glucose tolerance and lipid clearance, severe inflammation accompanied by dysfunctional liver and kidney all reduce the quality of life and lifespan in the elderly. In addition to the beneficial effects of adiponectin on healthspan and lifespan, this study also provides some insights into the adiponectin paradox and adiponectin resistance (*Zhao et al., 2021*). Some clinical observations imply the possible existence of an 'adiponectin paradox' in different disease settings (*Takamatsu et al., 2021*) (arguing that a paradoxical increase of all-cause and cardiovascular mortality can be observed with increased adiponectin levels), our study in aged mice with either increasing or reducing adiponectin levels should offer a clear answer to the existence of this 'adiponectin paradox'. If such an adiponectin paradox exists, chronically increasing adiponectin levels would result in a loss of its beneficial effects in aged mice. At least within the physiological range that we are operating on, this is clearly rejected by our current observations. In addition, with the results generated in our APN-KO mice, we would like to suggest that increasing adiponectin levels are always associated with beneficial effects in both young and old mice. Moreover, the chronic elevation of adiponectin levels in our transgenic mice also eliminates the possible existence of adiponectin resistance (*Engin, 2017*), as persistent beneficial effects are maintained in aged transgenic adiponectin mice. In addition, the most striking observations relate to the lack of adiponectin in aged mice, as the mice lacking adiponectin display a greatly increased risk of developing liver cancer. This completely contradicts a previous report claiming that adiponectin may be a promoting factor for hepatocellular carcinoma (*Chen et al., 2012*). Considering the robust effects of adiponectin in reducing inflammation and fibrosis in aged mice, we have strong reason to believe that adiponectin is indeed a protective factor to prevent the development of liver cancer. Further studies will be warranted to explore the mechanistic aspects of adiponectin effects leading to preventing liver cancer. Thus, the ability to prolong healthspan by maintaining adiponectin levels provides a promising therapy for aging-related disorders and improving quality of life in older individuals.

## Materials and methods

**Key resources table**

| Reagent type (species) or resource | Designation | Source or reference | Identifiers | Additional information |
|---|---|---|---|---|
| Genetic reagent (*Mus musculus*) | WT C57BL/6J | Jackson Laboratory | JAX 000664 RRID:IMSR_JAX:000664 | N/A |
| Genetic reagent (*Mus musculus*) | APN-KO | PMID:16326714 | N/A | N/A |
| Genetic reagent (*Mus musculus*) | ΔGly | PMID:14576179 | N/A | N/A |
| Chemical compound, drug | TRIzolTM Reagent | Thermo Fisher | Cat# 15596018 | N/A |
| Chemical compound, drug | Insulin | Eli Lilly | Product ID: A10008415 | N/A |
| Chemical compound, drug | Dulbecco's phosphate buffered saline | Sigma-Aldrich | Cat# D806552 | N/A |

*Continued on next page*

*Continued*

| Reagent type (species) or resource | Designation | Source or reference | Identifiers | Additional information |
|---|---|---|---|---|
| Chemical compound, drug | High-fat diet (HFD) | Research Diets | Cat# D12492 | N/A |
| Chemical compound, drug | DAPI | Life Technology | Cat# P36941 | N/A |
| Chemical compound, drug | Bovine serum albumin | Sigma | Cat# A9418 | N/A |
| Commercial assay or kit | Adiponectin ELISA kit | Invitrogen | Cat# EZMADP-60K RRID:AB_2651034 | N/A |
| Commercial assay or kit | Insulin ELISA Jumbo kit | ALPCO | Cat# 80-INSMS-E10 | N/A |
| Commercial assay or kit | Mouse/rat IGF-1 Quantikine ELISA kit | R and D | R and D Systems, Inc, Minneapolis, MN | N/A |
| Commercial assay or kit | Corticosterone Competitive ELISA kit | Invitrogen | Cat# EIACORT | N/A |
| Commercial assay or kit | iScript cDNA synthesis kit | BIO-RAD | Cat# 170–8891 | N/A |
| Commercial assay or kit | Sybr Green Master Mix | Thermo Fisher | Cat# A25778 | N/A |
| Commercial assay or kit | Senescence detection kit | Abcom | Cat#: AB65351 | N/A |
| Antibody | Mac2 (rat monoclonal) | BioLegend | Cat# 125401 RRID:AB_1134237 | IF(1:500) IHC(1:500) |
| Antibody | Perilipin (goat polyclonal) | Novus | Cat# NB100-60554 RRID:AB_922242 | IF(1:500) |
| Antibody | Insulin (guinea pig polyclonal) | Dako | Cat# A0564 RRID:AB_10013624 | IF(1:500) |
| Antibody | Glucagon (rabbit polyclonal) | Invitrogen | Cat# 15954–1-AP RRID:AB_2878200 | IF(1:500) |
| Antibody | Alexa Fluor 488 goat anti-guinea pig IgG (HCL) | Invitrogen | Cat# A-11073 RRID:AB_2534117 | IF(1:250) |
| Antibody | Alexa Fluor 594 donkey anti-rabbit IgG (HCL) | Invitrogen | Cat# A32754 RRID:AB_2762827 | IF(1:250) |
| Antibody | Alexa Fluor 594 donkey anti-goat IgG (HCL) | Invitrogen | Cat# A32758 RRID:AB_2762828 | IF(1:250) |
| Antibody | Alexa Fluor 488 goat anti-rat IgG (HCL) | Invitrogen | Cat# A48262 | IF(1:250) |
| Antibody | CD45-PerCP/Cyanine5.5 (rat monoclonal) | Biolegend | Cat# 103132 RRID:AB_893340 | FACS(1:400) |
| Antibody | CD11b-Pacific Blue (rat monoclonal) | Biolegend | Cat# 101224 RRID:AB_755986 | FACS(1:200) |
| Antibody | F4/80 -PE (rat monoclonal) | Biolegend | Cat# 123110 RRID:AB_893486 | FACS(1:200) |
| Antibody | CD11c -APC (Armenian Hamster monoclonal) | Biolegend | Cat# 117310 RRID:AB_313779 | FACS(1:200) |
| Antibody | CD206 -FITC (rat monoclonal) | Biolegend | Cat# 141703 RRID:AB_10900988 | FACS(1:200) |
| Sequence-based reagent | Gapdh _F | This paper | PCR primers | TGTGAACGGATTTGGCCGTA |
| Sequence-based reagent | Gapdh _R | This paper | PCR primers | ACTGTGCCGTTGAATTTGCC |

*Continued*

| Reagent type (species) or resource | Designation | Source or reference | Identifiers | Additional information |
|---|---|---|---|---|
| Sequence-based reagent | F4/80_F | This paper | PCR primers | TGACTCACCTTGTGGTCCTAA |
| Sequence-based reagent | F4/80_R | This paper | PCR primers | CTTCCCAGAATCCAGTCTTTCC |
| Sequence-based reagent | IL-6_F | This paper | PCR primers | CCAGAGATACAAAGAAATGATGG |
| Sequence-based reagent | IL-6_R | This paper | PCR primers | ACTCCAGAAGACCAGAGGAAAT |
| Sequence-based reagent | TNF$\alpha$_F | This paper | PCR primers | GAGAAAGTCAACCTCCTCTCTG |
| Sequence-based reagent | TNF$\alpha$_R | This paper | PCR primers | GAAGACTCCTCCCAGGTATATG |
| Software, algorithm | Prism | GraphPad Software | GraphPad Software | N/A |
| Software, algorithm | Illustrator | Adobe | N/A | N/A |
| Other | Keyence BZ-X700 fluorescence microscope | Keyence | N/A | N/A |
| Other | Zeiss Axioskop FS2 microscope | Zeiss | N/A | N/A |

## Animal experiments

APN-KO mice (*Nawrocki et al., 2006*) and ΔGly mice (*Combs et al., 2004*) with WT controls are on a pure C57BL/6J background. The transgenic strategy was to elevate adiponectin in the circulation using a deletion mutant of adiponectin under control of the adipose-specific enhancer/promoter of the aP2 gene. All of the animal experimental protocols have been approved by the Institutional Animal Care and Use Committee of University of Texas Southwestern Medical Center at Dallas. The mice were housed under standard laboratory conditions (12 hr on/off; lights on at 7:00 a.m.) and temperature-controlled environment with food and water available ad libitum. Mice were fed a standard NCD (number 5058, LabDiet, St. Louis, MO) or HFD (60% energy from fat, D12492, Research Diets) for various periods as indicated in the figures. All experiments were initiated at approximately 8 weeks of age, unless indicated otherwise. Mouse phenotyping studies were performed with controls and a minimum of two independent cohorts with more than five mice in each group.

## Systemic tests

Systemic tests were previously described (*Zhao et al., 2014*; *Zhu et al., 2017*). In brief, OGTTs were performed on overnight fasted mice. The mice orally received 1.25 or 2 g of glucose per kg body weight dissolved in phosphate buffered saline (Cat# 806552, Sigma-Aldrich). Injection volume was calculated based on 10 $\mu$l g$^{-1}$ body weight. Blood glucose concentrations were measured by glucose meters (Contour) at the indicated time points. For ITTs, mice were fasted for 6 hr in the morning, and chow-fed animals were intraperitoneally injected with insulin at a dose of 0.5 U kg$^{-1}$ body weight, while HFD-fed animals were injected with a dose of 0.75 U kg$^{-1}$ body weight. Blood glucose concentrations were measured by glucose meter at the indicated time points. For TG clearance, mice were fasted (16 hr), then gavaged 15 $\mu$l g$^{-1}$ body weight of 20% intralipid (Fresenius Kabi Clayton, L.P.). Blood was collected at timed intervals, then assayed for TG levels (Infinity, Thermo Fisher Scientific) and FFA levels (NEFA-HR, Wako Pure Chemical Industries). For some of the experiments, area under curve was calculated.

## Blood parameters

Blood was taken from fed animals in the morning and was centrifuged at 8000 g for 5 min, and then the supernatants were collected for multiple analyses. Adiponectin was measured using an ELISA kit from Invitrogen (Cat# EZMADP-60K). Serum insulin levels were measured using ALPCO Mouse Insulin ELISA Jumbo kit (Cat# 80-INSMS-E10, Mercodia Developing Diagnostic). Serum IGF-1 levels were measured by Mouse/Rat IGF-1 Quantikine ELISA kit (R and D Systems, Inc, Minneapolis, MN).

Corticosterone was measured using a Corticosterone Competitive ELISA Kit (Cat# EIACORT); serum parameters were measured and calculated with a VITROS analyzer (Ortho Clinical Diagnostics) at UTSW metabolic core.

## RT-qPCR and analysis

RNA was extracted from fresh or frozen tissues by homogenization in TRIzol reagent (Invitrogen) as previously described (*Zhu et al., 2016*). We used 1 µg RNA to transcribe cDNA with a reverse transcription kit (Bio-Rad). Most of RT-qPCR primers were from the Harvard Primer Bank (https://pga.mgh.harvard.edu/primerbank/). The relative expression levels were calculated using the comparative threshold cycle method, normalized to the housekeeping gene *Gapdh*.

## Histological analysis

For all histological analyses, four sections from at least three mice per group were stained and the examiner, typically a pathologist, was blinded to the genotype and/or treatment condition, as previously described (*Zhao et al., 2020*). In brief, for immunohistochemistry (IHC), tissues were fixed in 4% paraformaldehyde and embedded in paraffin. Sections (5 µm) were deparaffinized, heat retrieved (buffer with 10 mM Tris, 1.0 mM EDTA, pH = 8.0, 94–96℃ for 30 min, cool naturally), perforated (0.2% Triton × 100, 10 min), blocked in 3% BSA (Sigma, A9418) and then incubated with Mac2 (1:500 dilution, Cat# 125401, BioLegend) primary antibodies. IHC and Hematoxylin (Vector, H3401) and Eosin Y (Thermo, 6766007) staining (H&E staining) were performed using standard protocols or under the manufacturer's instructions. Detection of IHC signal was performed with Vectastain Elite ABC kit (Vector Laboratories, Burlingame, CA) and DAB substrate kit for peroxidase (Vector Laboratories) followed by hematoxylin counterstaining (Vector Laboratories). For immunofluorescence of perilipin (1:500 dilution NB100-60554, Novus), Mac2, insulin (1:500, Dako #A0564) and glucagon (1:500, Invitrogen #15954–1-AP), after incubation with primary antibody, slides were washed and incubated with secondary antibodies (1:250 dilution) used were Alexa Fluor 488 or 594 donkey anti-rabbit IgG (HCL), Alexa Fluor 488 or 594 donkey anti-goat IgG (HCL) (Invitrogen) or Alexa Fluor 488 or 594 donkey anti-guinea pig IgG (HCL) at room temperature for 1 hr, then washed and sealed with Prolong Gold antifade reagent with DAPI (Life Technology P36941). Picrosirius red and trichrome staining were performed by the histology core at UT Southwestern Medical Center.

## Metabolic cage experiments

Metabolic cage studies were conducted using a PhenoMaster System (TSE Systems) at USTW Metabolic Phenotyping Core as previously described (*Zhao et al., 2019*). Mice were acclimated in temporary holding cages for 5 days before recording. Food intake, movement, and $CO_2$ and $O_2$ levels were measured at various intervals (determined by collectively how many cages were running concurrently) for the indicated period shown on figures.

## Isolation of liver macrophages and flow cytometry

The detailed procedure used to isolate liver macrophages and flow cytometry has been recently described in our recent publication (*YA et al., 2021*). In brief, mouse livers from APN-KO and littermate control mice were perfused with digestion buffer. Then the digested mixture was centrifuged at low speed (50 g). After collecting the supernatant, neural progenitor cell (NPC) was harvested after centrifuge. Then the cells were resuspended in blocking buffer (2% FBS/PBS containing antimouse CD16/CD32 Fc Block 1:200). For macrophage flow cytometry and sorting, primary antibodies were supplemented into the cells in blocking buffer for 15 min incubation at 4℃. The cells were then washed once and resuspended in 2% FBS/PBS before sorted by a BD Biosciences FACSAria cytometer (BD, San Jose, CA) at the Flow Cytometry Core Facility at UT Southwestern. The primary antibodies and the working concentrations are as follows: CD45-PerCP/Cyanine5.5 1:400 (Biolegend, clone 30-F11, #103132), CD11b-Pacific Blue 1:200 (Biolegend, clone M1/70, #101224), F4/80-PE (Biolegend, clone BM8, #123110), CD11c-APC 1:200 (Biolegend, clone N418, #117310), CD206-FITC 1:200 (Biolegend, clone C068C2, #124808). Total RNA from freshly FACS-sorted macrophages was extracted and purified using the RNAqueous Micro total RNA isolation kit (Thermo Fisher Scientific, #AM1931). Subsequently, cDNA was synthesized using the random hexamer primers (Thermo Fisher Scientific, #N8080127) and M-MLV reverse transcriptase (Thermo Fisher Scientific, #28025013). All

the flow cytometric results were analyzed, and plots were generated with FlowJo (Version 10, FlowJo, BD). Total, pro- and anti-inflammatory macrophages in the liver are labeled as: total: CD45+/F4/80+/CD11b+; pro: pro-inflammatory macrophages (M1), CD45+/F4/80+/CD11b+/CD11c+/CD206-; anti: anti-inflammatory macrophages (M2), CD45+/F4/80+/CD11b+/CD11c-/CD206+.

### β-Galactosidase staining
Aged APN-KO and littermate control mice were used in this experiment. The mice were perfused through the heart initially by PBS, followed with 4% PFA. Kidney and liver were collected, frozen-sectioned, and stored at −20℃ for future use. β-Galactosidase staining was done using Senescence Detection Kit (Cat# ab65351). In brief, the slides were incubated with PBS for 15 min and fixed with fixative solution provided inside the kit for 15 min at room temperature. After washing the slides with PBS for two times, the slides were stained with staining buffer for overnight at 37℃. Followed with two times PBS wash, the slides were fixed with mounting media.

### Body composition and bone analyses
Body fat and lean mass were measured using an EchoMRI-100 (Echo Medical Systems, LLC). Bone mass was determined by DEXA scanning, as previously described (*Ford et al., 2011*).

### Statistics
All values are expressed as the mean ± SEM. The significance between the mean values for each study was evaluated by Student's *t* tests for comparisons of two groups. One-way or two-way ANOVA was used for comparisons of more than two groups. The box-and-whisker analysis was performed to exclude potential outliner data accordingly.

p≤0.05 is regarded as statistically significant. For lifespan analysis, data were calculated using the GraphPad Prism 7 and OASIS 2 software. Log-rank (Mantel–Cox) tests were used to analyze Kaplan–Meier curves. All the detailed statistical methods, sample sizes, and p-values are listed in the supplementary file.

### Study approval
The Institutional Animal Care and Use Committee of the University of Texas Southwestern Medical Center approved all animal experiments (APN:2015–101207G).

## Acknowledgements
We thank all members of Scherer for their support of this study. We would also like to thank the UTSW Metabolic Core Facility, the Histo-Pathology Core, UTSW ARC, and Charlotte Lee for help with histology. Funding: This study was supported by US National Institutes of Health grant P01-AG051459 to TLH, VDD, and PES.

## Additional information

### Funding

| Funder | Grant reference number | Author |
|---|---|---|
| National Institutes of Health | P01-AG051459 | Tamas L Horvath Vishwa Deep Dixit Philipp E Scherer |
| National Institutes of Health | K01-DK125447 | Yu A An |

The funders had no role in study design, data collection and interpretation, or the decision to submit the work for publication.

### Author contributions
Na Li, Formal analysis, Investigation, Methodology, Writing - original draft; Shangang Zhao, Investigation, Methodology, Writing - review and editing; Zhuzhen Zhang, Conceptualization, Investigation, Methodology; Yi Zhu, Investigation, Writing - review and editing; Christy M Gliniak, Writing - review

and editing; Lavanya Vishvanath, Bo Shan, Amber Sherwood, Methodology; Yu A An, Tamas L Horvath, Conceptualization, Investigation; May-yun Wang, Formal analysis, Methodology; Yingfeng Deng, Resources, Investigation; Qingzhang Zhu, Conceptualization; Toshiharu Onodera, Resources; Orhan K Oz, Methodology, Writing - review and editing; Ruth Gordillo, Resources, Methodology; Rana K Gupta, Conceptualization, Methodology; Ming Liu, Vishwa Deep Dixit, Conceptualization, Writing - review and editing; Philipp E Scherer, Conceptualization, Supervision, Funding acquisition, Writing - review and editing

**Author ORCIDs**
Na Li ![ORCID] https://orcid.org/0000-0002-0604-6312
Shangang Zhao ![ORCID] https://orcid.org/0000-0002-9209-8206
Zhuzhen Zhang ![ORCID] https://orcid.org/0000-0001-6787-3920
Yu A An ![ORCID] https://orcid.org/0000-0002-9678-3382
Yingfeng Deng ![ORCID] http://orcid.org/0000-0003-1314-5105
Toshiharu Onodera ![ORCID] https://orcid.org/0000-0002-4439-0077
Rana K Gupta ![ORCID] http://orcid.org/0000-0002-9001-4531
Ming Liu ![ORCID] https://orcid.org/0000-0003-2665-4072
Tamas L Horvath ![ORCID] http://orcid.org/0000-0002-7522-4602
Vishwa Deep Dixit ![ORCID] http://orcid.org/0000-0002-5341-6494
Philipp E Scherer ![ORCID] https://orcid.org/0000-0003-0680-3392

**Decision letter and Author response**
Decision letter https://doi.org/10.7554/eLife.65108.sa1
Author response https://doi.org/10.7554/eLife.65108.sa2

## Additional files

### Supplementary files
• Source data 1. All raw datasets of each figure for this study.

• Supplementary file 1. Statistical information in each figure. Table displaying the sample size, statistical test method, and p-value for the list figures.

• Transparent reporting form

### Data availability
All data generated or analysed during this study are included in the manuscript and supporting files.

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
