## [Decision Letter]

**Acceptance summary:**

This interesting manuscript better defines a role for adiponectin in inflammation and aging using loss and gain of function mice. This provides important new insights and novel therapeutic targets.

**Decision letter after peer review:**

Thank you for submitting your article "Adiponectin Preserves Metabolic Fitness During Aging" for consideration by *eLife*. Your article has been reviewed by 3 peer reviewers, including Carlos Isales as the Reviewing Editor and Reviewer #1, and the evaluation has been overseen by a Mone Zaidi as the Senior Editor.

Summary:

Adiponectin is a key adipokine, and much of our knowledge about this molecule has come from the Scherer lab. It is well known that adiponectin promotes improved insulin sensitivity and glucose tolerance, along with anti-inflammatory effects, which can be followed by decreased fibrosis. In this paper the Authors use loss and gain of function mouse models to explore whether the beneficial effects of adiponectin on metabolism can be translated into greater healthspan or lifespan. They show that lifespan decreases in adiponectin KOs and increases in the transgenic (ΔGly) mice. The expected effects on glucose metabolism, insulin sensitivity, inflammation, and fibrosis are also demonstrated.

Essential revisions:

1. Given the known significant effects of adiponectin on metabolic fitness, the effects on healthspan which the Authors observed in their 2 models, was expected. However, while median survival time is definitely less in the APN-KOs and greater in the ΔGly mice, the effects are relatively modest compared to other longevity studies. Any increase in lifespan is a good thing, particularly when accompanied by a corresponding increase in healthspan. We would've hoped for greater effects on lifespan than those observed but even modest effects are worthwhile. The Authors should comment in their discussion on this point. In other words, it would be good to know the Authors' thinking as to why these impressive effects on glucose, insulin, inflammation, fibrosis, etc. do not lead to even greater effects on lifespan. Also, is there any information on the causes of mortality in the WT vs. KOs that might point to why lifespan is decreased?

2. APN-KO clearly leads to impaired glucose tolerance, but it is a bit surprising why insulin levels aren't increased, which is the typical metabolic response to insulin resistance.

3. Can the Authors please comment on adipose tissue mass in the KOs, particularly if they have any information on subq fat?

4. In Figure 3, they show increased staining for ATMs with Mac2 in the KOs. What about the expression of other inflammatory gene markers, such as those shown in Figure 3G for the liver?

5. With respect to hepatic effects, this paper shows increased inflammation in the liver in APN-KOs. However these gene expression patterns are in total liver tissue, and it would be helpful to understand the origin of these inflammatory markers. Are they from Kupffer cells, monocyte-derived macrophages, etc. In a similar vein, various fibrosis marker genes are increased in total liver from the APN-KOs. Most likely these expression differences reflect stellate cell effects. Do the Authors have any information on the effect of adiponectin on stellate cell function. Although fibrosis-related genes are elevated in the APN-KO, is there histologic evidence of increased fibrosis in the liver sections?

6. The Authors suggest that the increased inflammation in the liver is the cause of the increased fibrosis. Presumably they think that the immune cells in the liver are signaling to stellate cells to produce this effect. Is this the scenario the Authors propose. If so, it should be made more explicit and corroborated by histologic staining of hepatic fibrosis.

7. It would be of interest to know the extent of inflammation in the kidneys with APN-KO, beyond Mac2 staining (Figure 3D).

8. In the results in the ΔGly mice, is the enhanced lifespan statistically significant. Unless we are misreading it, the p value suggests it is not. Also, why have only study chow fed mice and not HFD mice in the transgenics, as they did in KOs?

9. ITTs are shown in Figure 4G, but the basal glucose values are different between the 2 groups. Can the Authors also present the data normalized to the basal value to determine whether the kinetics of the curve are different?

10. The resulting changes in tissue fibrosis are clearly important when thinking about healthy tissue function. It would help if the authors could show histologic staining for collagen deposition in the various tissues, particularly liver and kidney. Although it might be asking for too much if the they don't already have this information, it would also be useful to know which cell types within the various tissues are responsible for the changes in inflammatory markers and collagen related genes. This could also be discussed.

11. From an aesthetic point of view there is a certain lack symmetry in this paper, since some of the measurements made in the KOs are not performed in the transgenics and HFD was not utilized in the transgenics either.

12. Much of the data could be predicted from studies by them or the other investigators in the field (Nature Med. 8, 731 [2002], J. Biol. Chem. 277, 25863 [2002], J. Biol. Chem. 277, 34658 [2002], J. Biol. Chem. 278, 2461 [2003], Endocrinology 145, 367 [2004], J. Biol. Chem. 281, 2654 [2006], Am. J. Physiol. Endocrinol. Metab. 293, 210 [2007], J. Clin. Invest. 118, 1645 [2008]). IT would be helpful if authors could provide insights into the life-promoting mechanism by adiponectin that has not been clarified so far.

---

## [Author Response]

Essential revisions:1. Given the known significant effects of adiponectin on metabolic fitness, the effects on healthspan which the Authors observed in their 2 models, was expected. However, while median survival time is definitely less in the APN-KOs and greater in the ΔGly mice, the effects are relatively modest compared to other longevity studies. Any increase in lifespan is a good thing, particularly when accompanied by a corresponding increase in healthspan. We would've hoped for greater effects on lifespan than those observed but even modest effects are worthwhile. The Authors should comment in their discussion on this point. In other words, it would be good to know the Authors' thinking as to why these impressive effects on glucose, insulin, inflammation, fibrosis, etc. do not lead to even greater effects on lifespan. Also, is there any information on the causes of mortality in the WT vs. KOs that might point to why lifespan is decreased?

As suggested by reviewers, one more paragraph has been added in the discussion as follows:

Regarding the effect of adiponectin in lifespan, it is quite clear that APN-KO mice with no circulating adiponectin significantly shorten their lifespan, while adiponectin transgenic mice with an increase in circulating adiponectin only slightly increase their median lifespan, despite the substantial improvement in glucose and insulin tolerance and significant reduction in tissue inflammation and fibrosis. Tissue inflammation and fibrosis are suggested as one of the crucial contributing factors for lifespan. Consistent with this notion, the increased liver inflammation and fibrosis in APN-KO mice results in a higher incidence of liver cancer. In HFD-fed aged WT mice, we observed a small percentage of mice with liver cancer (<10%). However, in HFD-fed aged APN-KO mice, the rate of liver cancer increased up to 60%. Thus, this increased rate of liver cancer and its associated cachexia possibly account for the shortened lifespan in APN-KO mice. In addition, this observation also suggests that inflammation or fibrosis itself is not the actual cause of death for the mice. This may help explain the reduced inflammation and fibrosis in adiponectin transgenic mice only slightly increases their lifespan, implicating other factors, in addition to reduced inflammation and fibrosis, may contribute to the natural death of these mice.

2. APN-KO clearly leads to impaired glucose tolerance, but it is a bit surprising why insulin levels aren't increased, which is the typical metabolic response to insulin resistance.

A very interesting question. A typical metabolic response to insulin resistance is to increase circulating insulin levels to overcome the insulin resistance. Thus, impaired glucose tolerance in mice with high degrees of insulin resistance is always associated with higher circulating insulin levels, especially under conditions of high fat diet. However, in our APN-KO mice, while we observed impaired glucose and insulin tolerance, the circulating insulin levels do not increase. This observation can at least be partially explained by the direct effects of adiponectin on β cell function. In one of our previous reports, we have demonstrated that adiponectin is essential to mitigate local lipotoxicity in pancreatic islets and promote reconstitution of β cell mass. The lack of adiponectin in APN-KO mice directly impairs β cell function to compromise the cell’s ability to elevate insulin secretion, resulting in lower insulin levels in response to impaired glucose tolerance.

3. Can the Authors please comment on adipose tissue mass in the KOs, particularly if they have any information on subq fat?

A new figure (Figure 2—figure supplement 1) has been added in the revised version of this manuscript. We have added fat mass, lean mass, weights of subcutaneous, gonadal fat and brown fat.

4. In Figure 3, they show increased staining for ATMs with Mac2 in the KOs. What about the expression of other inflammatory gene markers, such as those shown in Figure 3G for the liver?

A new figure (Figure 4A) regarding inflammatory gene markers has been added in the revised version of this manuscript.

5. With respect to hepatic effects, this paper shows increased inflammation in the liver in APN-KOs. However these gene expression patterns are in total liver tissue, and it would be helpful to understand the origin of these inflammatory markers. Are they from Kupffer cells, monocyte-derived macrophages, etc. In a similar vein, various fibrosis marker genes are increased in total liver from the APN-KOs. Most likely these expression differences reflect stellate cell effects. Do the Authors have any information on the effect of adiponectin on stellate cell function. Although fibrosis-related genes are elevated in the APN-KO, is there histologic evidence of increased fibrosis in the liver sections?

An excellenet suggestion! As suggested, in the revised version, we have isolated macrophages from livers of WT and APNKO mice and sorted out Kupffer cells and monocyte-derived macrophages. We Figure 4C found that total macrophages in APNKO mice are increased by a factor of two (Figure 4C). Moreover, relative to WT mice, Kupffer cells and monocyte-derived macrophages both increased in APN-KO mice, and Kupffer cells are the major source of the increased inflammatory response (Figure 4C).

Regarding the effects of adiponectin on stellate cell function, a new manuscript is under preparation to specifically address this question.

In order to further corroborate the evidence for enhanced hepatic fibrosis, we have done additional trichrome and picrosirius red stains (Figure 3F), added in the revised version.

6. The Authors suggest that the increased inflammation in the liver is the cause of the increased fibrosis. Presumably they think that the immune cells in the liver are signaling to stellate cells to produce this effect. Is this the scenario the Authors propose. If so, it should be made more explicit and corroborated by histologic staining of hepatic fibrosis.

Another excellent point. We fully agree with this scenario. We have an entire manuscript under preparation dedicated to the effects of adiponectin on hepatic stellate cells which will be submitted soon. In the revised version, we have added one paragraph in the discussion:

“Hepatic stellate cells are the major cell types that produce collage in response to liver injury. In the APNKO mice, a significant increase in liver inflammation by elevated total macrophage number and enhanced activity provides a strong insult to liver, resulting in hepatic stellate cell activation and liver fibrosis. In contrast, higher circulating levels of adiponectin, observed in adiponectin transgenic mice, greatly reduce liver inflammation, leading to reduced liver fibrosis. All these results suggest that the local inflammation in the liver is the major driver of liver fibrosis.

7. It would be of interest to know the extent of inflammation in the kidneys with APN-KO, beyond Mac2 staining (Figure 3D).

Thank you for the suggestion. In the revised version, we have added gene expression of inflammatory markers in the kidneys with APN-KO mice (Figure 4B).

8. In the results in the ΔGly mice, is the enhanced lifespan statistically significant. Unless we are misreading it, the p value suggests it is not. Also, why have only study chow fed mice and not HFD mice in the transgenics, as they did in KOs?

As suggested by p value, the maximum lifespan does not reach statistical significance. However, the median lifespan in the ΔGly mice is greatly increased.

In this experiment, we did not perform HFD study in the ΔGly mice, due to the nature of aging study with large number of mice. For some unknown reasons, ΔGly mice do not breed very well, with fewer resulting offspring. We had a very difficult time to collect enough mice to initiate our aging study. So we decided to only perform a chow study in the ΔGly mice.

9. ITTs are shown in Figure 4G, but the basal glucose values are different between the 2 groups. Can the Authors also present the data normalized to the basal value to determine whether the kinetics of the curve are different?

As suggested by the reviewer, a new figure (Figure 5G) has been added in the revised version.

10. The resulting changes in tissue fibrosis are clearly important when thinking about healthy tissue function. It would help if the authors could show histologic staining for collagen deposition in the various tissues, particularly liver and kidney. Although it might be asking for too much if the they don't already have this information, it would also be useful to know which cell types within the various tissues are responsible for the changes in inflammatory markers and collagen related genes. This could also be discussed.

We have performed picrosirius red staining in the kidney and liver, added in the revised version (Figure 4F). In addition, with FACS-sorted Kupffer cells and monocyte-derived macrophages, we found that Kupffer cells are the major sources of the inflammatory response (Figure 4C).

A new paragraph has been added in the revised version.

11. From an aesthetic point of view there is a certain lack symmetry in this paper, since some of the measurements made in the KOs are not performed in the transgenics and HFD was not utilized in the transgenics either.

We understand and appreciate this point. We strived to achieve symmetry in our manuscript, as the majority of measurements have been performed both in the KO and transgenic mice. However, due to the nature of aging studies with large number of transgenic mice, we were unable to collect a sufficient number of mice to initiate aging studies on both chow and HFD diets at the same time. This is the major reason for the asymmetries, and we appreciate that as a shortcoming.

12. Much of the data could be predicted from studies by them or the other investigators in the field (Nature Med. 8, 731 [2002], J. Biol. Chem. 277, 25863 [2002], J. Biol. Chem. 277, 34658 [2002], J. Biol. Chem. 278, 2461 [2003], Endocrinology 145, 367 [2004], J. Biol. Chem. 281, 2654 [2006], Am. J. Physiol. Endocrinol. Metab. 293, 210 [2007], J. Clin. Invest. 118, 1645 [2008]). IT would be helpful if authors could provide insights into the life-promoting mechanism by adiponectin that has not been clarified so far.

Predictable – yes. However, these studies have never been performed systematically in an actual aging study. As a response, we have added one more paragraph discussion in the revised version.

“In addition to the beneficial effects of adiponectin on healthspan and lifespan, this study also provides some insights into the adiponectin paradox and adiponectin resistance (Zhao, Kusminski et al., 2021). […] Further studies will be warranted to explore the mechanistic aspects of adiponectin effects leading to preventing liver cancer”